# Harnessing the Immune System: Current and Emerging Immunotherapy Strategies for Pediatric Acute Lymphoblastic Leukemia

**DOI:** 10.3390/biomedicines11071886

**Published:** 2023-07-03

**Authors:** Chana L. Glasser, Jing Chen

**Affiliations:** 1Department of Pediatric Hematology/Oncology, NYU Langone Hospital, Mineola, NY 11501, USA; 2Department of Pediatric Hematology/Oncology, Hackensack University Medical Center, Hackensack, NJ 07601, USA

**Keywords:** acute lymphoblastic leukemia, pediatric, children, immunotherapy, novel therapy

## Abstract

Treatment for relapsed acute lymphoblastic leukemia (ALL) in children and young adults continues to evolve. Despite optimization of cytotoxic chemotherapeutic approaches and risk-adapted therapy, about 12% of pediatric patients still relapse, and survival rates in this population remain poor. Salvage therapy for relapsed patients continues to be challenging as attempts to further intensify chemotherapy have resulted in excessive toxicity without improving outcomes. Immunotherapy has profoundly impacted the landscape of relapsed ALL by harnessing the patient’s immune system to target and eliminate leukemia cells. In this review, we provide an overview and summary of immunotherapy agents that have been approved and remain under investigation for children, including blinatumomab, inotuzumab, daratumomab, and chimeric antigen receptor T-cell therapy. We discuss the landmark clinical trials that have revolutionized the field and provide an update on ongoing clinical trials involving these agents for children in the relapsed and upfront setting. The incorporation of these novel immunotherapies into ALL treatment, either as monotherapy or in combination with cytotoxic chemotherapy, has demonstrated promising potential to augment outcomes while decreasing toxicity. However, we also highlight the many challenges we still face and the research critically needed to achieve our goals for cure in children.

## 1. Introduction

Treatment for relapsed B-acute lymphoblastic leukemia (ALL) and T-ALL continues to evolve. Despite optimization of multi-agent chemotherapy regimens, risk stratification, MRD monitoring, CNS prophylaxis, and supportive care measures, about 12% of children with acute lymphoblastic leukemia still relapse with a 5 yr post-relapse overall survival (OS) for B-ALL and T-ALL of 52% and 33%, respectively [1,2]. Therapy for these high-risk, relapsed patients continues to be challenging as attempts to further intensify chemotherapy has resulted in excessive toxicity without improving outcomes. Immunotherapy has profoundly impacted the landscape of relapsed ALL by harnessing the patient’s own immune system to target and eliminate leukemia cells while also reducing overall toxicity of anti-leukemic treatment. Several immunotherapy approaches for ALL have been established and others remain under development. Antibody-based therapy consists of anti-leukemic agents designed to direct the body’s immune system to recognize surface antigens on leukemic blast cells and eliminate them. These include bispecific T-cell engagers (BiTEs), such as blinatumomab; antibody-drug conjugates, such as inotuzumab (InO); and monoclonal antibodies, such as daratumumab. Chimeric antigen receptor (CAR T) cell therapy is a T-cell-based immunotherapy that has revolutionized the field of relapsed/refractory ALL therapy, offering potential cure for what was considered in many cases to be an incurable disease. The incorporation of these novel immunotherapy approaches into ALL therapy, either as monotherapy or in combination with cytotoxic chemotherapy regimens, continues to be investigated in clinical trials, and, while we have made tremendous progress, we continue to face many challenges. The goal of this review is to provide an overview of novel immunotherapies for treatment of ALL in children and to summarize the landmark clinical trials that have been published and those that are ongoing for this population.

### 1.1. B-ALL

#### 1.1.1. Blinatumomab

Blinatumomab (Blincyto, Amgen, Jung-so, Seoul) is a first-in-class recombinant murine protein that acts as a bispecific CD19-directed CD3 T-cell engager (BiTE). CD19 is an early B-lineage restricted antigen expressed in >90% of leukemic blasts and thereby an attractive drug target. Blinatumomab simultaneously binds CD3-positive cytotoxic T cells and CD19-positive B cells, resulting in T-cell-mediated serial lysis of normal and malignant B cells [3,4] (Figure 1). Blinatumomab was first FDA-approved in 2014 for Philadelphia (Ph) chromosome-negative relapsed/refractory (R/R) B-ALL in adults [5]. The approval was updated in 2017 to include Ph+ B-ALL and expanded in 2018 to include both children and adults in first or second CR with minimal residual disease (MRD) ≥ 0.1% [6,7].

##### Blinatumomab Adult Data

In 2014, blinatumomab was granted accelerated FDA approval for adults with Ph-negative R/R B-ALL based on a multicenter single arm Phase II study, in which 189 patients were treated with blinatumomab (9 μg/day × 7 days and 28 μg/day thereafter continuous infusion for 28 days) every 6 weeks for up to 5 cycles. Outcomes revealed a high complete response (CR)/CR with incomplete hematologic recovery (CRh) (defined as the disappearance of all signs of leukemia in response to treatment with residual neutropenia (less than 1000 per microliter), with or without complete platelet recovery) rate of 43% after 2 cycles, 82% of whom were minimal residual disease (MRD)-negative [8,9].

Full FDA approval was granted in 2017 and extended to include both Ph-negative and Ph-positive R/R B-ALL in adults based on the TOWER and ALCANTRA clinical trials. TOWER was a Phase III international randomized open label study of blinatumomab versus standard-of-care in adults with Ph-negative R/R B-ALL, in which outcome measures by complete hematologic remission, MRD response, event free survival (EFS), and overall survival (OS) consistently favored blinatumomab over chemotherapy. Patients who received blinatumomab experienced CR rates double those who received chemotherapy (34% vs. 16%, *p* < 0.001) as well as a twofold improvement in median OS (7.7 months versus 4 months) [10]. ALCANTRA was a single-arm Phase II trial of blinatumomab in Ph-positive R/R B-ALL which showed an improvement in outcomes from historical controls [11].

In 2018, blinatumomab received a second FDA-approved indication for treatment of adults with newly diagnosed B-ALL in CR with persistent positive MRD ≥ 0.1% after receiving conventional chemotherapy. This approval was based on results from the open label multicenter, single-arm trial (BLAST) in which 88 of 113 evaluable patients (78%) achieved MRD negativity after one cycle of blinatumomab with a relapse-free survival (RFS) of 54% at 18 months in the Ph-negative sub-population (n = 110) [12]. These results confirmed and expanded those of the initial pilot study in which 80% of patients achieved an MRD-negative response and the RFS at 33 months was 61% [13].

##### Blinatumomab Pediatric Data (Table 1)

FDA approval in adults laid the groundwork for investigation of blinatumomab in the pediatric population. The first pivotal study in children was an international multi-center Phase I/II study of single-agent blinatumomab in children with R/R B-ALL (including relapse post-HSCT) conducted in 26 European and US Centers by the International- Berlin–Frankfurt–Münster (I-BFM) consortium (NCT01471782). Forty-nine patients were treated in the Phase I portion and forty-four in Phase II. Four patients had DLTs including four with grade 4 CRS and one with fatal respiratory failure. Nine (13%) experienced a neurologic event, but all were grade 2 and resolved. The recommended Phase II dose (RP2D) was 5 mcg/m^2^/day × 7 days with escalation to 15 mcg/m^2^/day thereafter for a total of 28 days. Among the 70 patients who received the recommended dose, 39% achieved CR after 2 cycles, 52% of whom were MRD negative. The CR rate for relapsed only patients (excluding refractory) was 48%. The most frequent AEs were consistent with prior studies, including anemia (36%), thrombocytopenia (21%), hypokalemia (17%), and grade 3–4 cytokine release syndrome (CRS) (5%) [14]. When outcomes were compared to three historical comparator groups (North America, Australia, and Europe) treated with standard-of-care (SOC) cytotoxic chemotherapy, single-agent blinatumomab was associated with longer overall survival (OS) and a trend for higher CR compared to SOC chemotherapy [15]. Based on the promising results of this study, accelerated FDA approval was granted to children with R/R B-ALL with >0.1% disease in 2018 [7].

**Table 1 biomedicines-11-01886-t001:** Pediatric Clinical Trials with Blinatumomab.

Patient Population Number of Patients (n) Age Range	Study Phase NCT# Design	Outcome	Toxicity	Refs.
R/R 93 0–17 years	I/II NCT01471782 (COG AALL1121) Monotherapy	Of 70 who received RP2D: CR 39% after 2 cycles, of whom 52% MRD-negative	Grade 4 CRS (n = 4) Grade 5 respiratory failure (n = 1) Grade 2 neuro (n = 9)	[14]
1st relapse HR/IR 208 1–30 years	III NCT02101853 (COG AALL1331) Chemo ± Blina	2-year DFS 54.4% (Blina) vs. 39% (chemo) (*p* = 0.03) 2-year OS 71.3% (Blina) vs. 58.4% (chemo) (*p* = 0.02)	Grade 3–4 toxicity (infection, sepsis, mucositis) higher in chemo arm Toxic deaths: Chemo (n = 5) Blina (none) Blina: ≥Grade 3 CRS (n = 1) Neuro (n = 5)	[16]
1st relapse LR 255 1–30 years	III NCT02101853 (COG AALL1331) Chemo ± Blina	No significant difference overall Subgroup BM +/− EM relapse: 4-year DFS 74% (Blina) vs. 51.8% (chemo) (*p* = 0.016) 4-year OS 96.6% (Blina) vs. 84.4% (chemo) (*p* = 0.013) Isolated CNS relapse: 61% 2nd relapse	≥Grade 3 toxicity significantly higher in chemo arm (% per cycle): CRS (all grades) 12/7/7 Neuro (all grades): Seizure 3/1/3 Other 19/9/5 All reversible	[17]
1st relapse 108 ≤18 years	III NCT02393859 Chemo ± Blina	MRD-negative CR: 90% (Blina) vs. 54% (chemo) 2-year EFS: 66.2% (Blina) vs. 27.1% (chemo) (*p* < 0.001)	Serious toxicity 24.1% (Blina) vs. 43.1% (chemo) ≥Grade 3 toxicity 57.4% (Blina) vs. 82.4% (chemo)	[18]
1st relapse 1–31 years	II NCT04546399 (COG AALL1821) HR: Blina ± Nivo + HSCT LR: Chemo + Blina ± Nivo	Ongoing		
Frontline-SR 1–10 years (Down Syndrome and Lymphoma) 1–31 years	III NCT03914625 Chemo ± Blina for subsets	Ongoing		
Frontline-MR and HR 0–17 years	III NCT03643276 Chemo ± Blina	Ongoing		

Subsequently, COG conducted a large randomized controlled trial (RCT) to evaluate post-reinduction consolidative therapy with blinatumomab versus cytotoxic chemotherapy for children and adolescents/young adults age 1–30 with first relapse of B-ALL (COG AALL1331) (NCT02101853). The study was risk-stratified with high-risk (HR)/intermediate-risk (IR) patients randomized to consolidative blinatumomab monotherapy versus cytotoxic chemotherapy followed by hematopoietic stem cell transplant (HSCT) and low-risk (LR) patients randomized to consolidative reduced cytotoxic chemotherapy with addition of blinatumomab versus chemotherapy alone (Table 2). In the HR/IR cohort, there was a trend toward improved survival in the blinatumomab group at a median follow up of 2.9 years, but it was not statistically significant (2-year disease-free survival (DFS) 54.4% for blinatumomab group versus 39.0% for chemotherapy group [1-sided *p* = 0.03] and 2-year OS 71.3% for blinatumomab group versus 58.4% for chemotherapy group [1-sided *p* = 0.02]). Rates of serious grade 3–4 adverse events included infection, febrile neutropenia, sepsis, and mucositis were significantly higher in the chemotherapy group, leading to early closure of the randomization [16].

In the LR cohort, while there was no significant difference in outcome for the entire population, the blinatumomab arm was superior to the standard chemotherapy arm for the sub-group of patients with bone marrow (BM) ± extramedullary (EM) relapse (4-year DFS was 74.0 ± 6.4% for blinatumomab vs. 51.8 ± 7.9% for chemotherapy [*p* = 0.016], and 4-year OS was 96.6 ± 2.5% for blinatumomab vs. 84.4 ± 5.6% for chemotherapy [*p* = 0.013]), establishing this regimen as a new standard of care for this sub-set of patients. However, of note, there was no benefit of blinatumomab observed for patients with late isolated EM relapse when compared to chemotherapy alone. Rather, there was an excess of second relapse in this population (61%), of which the majority were isolated central nervous system (CNS) relapses (72%), highlighting a major shortcoming of blinatumomab in penetrating and effectively treating CNS disease [17].

Simultaneously, the I-BFM consortium conducted a multicenter open-label, randomized, controlled IntReALL study for children ≤ 18 years old high risk first relapse B-ALL in morphologic CR (M1 or M2 marrow) evaluating blinatumomab versus chemotherapy for the third consolidation course (NCT02393859). A total of 108 patients were enrolled (54 in the blinatumomab group and 54 in the chemotherapy group). The study was terminated early due to a benefit of the blinatumomab meeting a pre-specified stopping rule. At a median of 22.4 months follow-up, the incidence of events in the blinatumomab arm vs. chemotherapy was 31% vs. 57% (*p* < 0.01), and the rate of MRD-negative remission was significantly higher in the blinatumomab arm (90% [44/49]) versus the chemotherapy arm (54% [26/48]). Furthermore, serious adverse events were significantly increased in the chemotherapy arm, as was observed on the COG study [18].

Building off this impressive efficacy and tolerability data, a successor COG Phase II risk-stratified study, AALL1821 (NCT04546399), was opened in December 2020 to assess whether the addition of the checkpoint inhibitor, nivolumab to blinatumomab would augment blinatumomab activity by improving endogenous T-cell activation and expansion, necessary for blinatumomab activity. This study, which further reduces exposure to cytotoxic chemotherapy from the SOC, is ongoing at this time.

Furthermore, with the FDA approval in children with persistent MRD and robust data in the relapse setting, blinatumomab was considered an ideal candidate to move forward to investigate in the frontline setting for children with newly diagnosed NCI-standard risk (SR) ALL. COG AALL1731 (NCT03914625), which opened in June 2019, is a Phase III risk-stratified RCT, in which children with SR ALL with particular unfavorable features, including neutral cytogenetics with end-of-induction (EOI) high throughput sequencing (HTS) MRD positivity, unfavorable cytogenetics, and high EOI MRD by flow cytometry, are randomized to receive the standard chemotherapy backbone alone versus with integration of two courses of blinatumomab. Children with high MRD at the end of consolidation are non-randomly assigned to receive blinatumomab. This exciting study is ongoing.

Blinatumomab is also being evaluated frontline internationally for intermediate and high risk patients on the AIEOP-BFM ALL 2017 study. On this study, intermediate risk patients, as defined by genetics and intermediate MRD response, are randomized to receive additional therapy with one cycle of post-reintensification blinatumomab versus SOC chemotherapy alone. High-risk patients, as defined by genetics and/or inadequate treatment response by the end of consolidation, are randomized to receive two cycles of post-consolidation immunotherapy with blinatumomab in place of two conventional highly intensive chemotherapy courses versus SOC chemotherapy. This important study is ongoing as well (NCT03643276).

#### 1.1.2. Inotuzumab

Inotuzumab ozogamicin (InO, Besponsa^®^, Pfizer, New York, NY, USA) is an antibody–drug conjugate composed of a humanized anti-CD22 monoclonal antibody conjugated to calicheamicin, a cytotoxic agent derived from *Micromonospora echinospora*. It binds with high affinity to CD22, a cell-surface glycoprotein antigen expressed by >90% of B-cell blasts and is involved in B-cell activation and regulation. Upon binding to CD22 on leukemic blasts, the antibody–drug conjugate is rapidly internalized into the cell, where calicheamicin is released, hydrolyzed and reduced into a reactive intermediate, which cleaves double-stranded DNA, resulting in cellular apoptosis and cell-cycle arrest [3,19] (Figure 1). Early phase clinical trials in both adults and children showed impressive efficacy in R/R patients, leading to incorporation of InO into upfront Phase III clinical trials and FDA approval in adults with R/R disease in 2017. Clinical trials in pediatrics for both relapse and frontline therapy are ongoing.

##### Inotuzumab Adult Data

The first report of InO in B-ALL came from a Phase II study that enrolled 49 adults and children at MD Anderson Cancer Center (MDACC), with median age 36 years (6–80 years). Patients were treated with single dose InO at 1.8 mg/m^2^ every 3–4 weeks, the dosing previously established for R/R non-Hodgkin lymphoma. The overall response rate was 57%. Nine patients (18%) had a CR, nineteen (39%) had marrow CR with incomplete peripheral count recovery, nineteen (39%) had resistant disease, and two (4%) died early in the study [20]. Subsequent studies, both single-center at MDACC and multi-institutional, established weekly fractionated dosing of InO (0.8 mg/m^2^ day 1 and 0.5 mg/m^2^ days 8 and 15) to be superior due to equivalent efficacy with decreased toxicity [21,22].

InO received FDA approval in 2017 for treatment of R/R B-ALL in adults based on the landmark INotuzumab Ozogamicin trial to investigate Tolerability and Efficacy (INO-VATE), an open label multinational Phase III RCT, comparing InO monotherapy to SOC intensive chemotherapy for R/R B-ALL in adults. Of the 307 patients randomized on INO-VATE, the InO arm had a significantly higher rate of CR/CR with incomplete hematologic recovery (Cri) compared to the chemotherapy arm (73.8% versus 30.9%, 1-sided *p* < 0.0001) and higher 2-year OS (22% versus 10%, 1-sided *p* = 0.0105). More patients on the InO arm proceeded to HSCT. The most common adverse events were hematologic in both arms, but the rate of veno-occlusive disease (VOD)/sinusoidal obstructive syndrome (SOS) was higher in the InO arm (23/164 [14%] versus 3/143 [2.1%]) [19,23].

With this approval, the Alliance A041501 trial (NCT03150693), an RCT evaluating the addition of InO to a modified augmented BFM chemotherapy backbone, opened in 2017 for young adults with newly diagnosed B-ALL. Unfortunately, the study was closed early due to unacceptable, Grade 5 toxicity in the InO arm, primarily related to sepsis in the setting of myelosuppression during chemotherapy blocks as well as to hepato-biliary and renal toxicities. While study results have not yet been published, this early closure raised concern about the safety of combining InO with chemotherapy in frontline therapy [24,25].

##### Inotuzumab Pediatric Data (Table 3)

The first published report of InO in children was a retrospective cohort study of 51 children ≤ 21 years old with heavily pre-treated ALL who received InO at the FDA-approved fractionated dosing in a compassionate use program across 30 centers worldwide. Of 42 patients with overt relapse (M2/M3 marrow), 67% achieved CR, 71% of whom were MRD-negative. InO was well tolerated, with notable toxicities including grade 3 hepatic toxicity in 6 patients (12%) and grade 3–4 infections in 11 (22%). No SOS was observed during therapy, but 11/21 patients (52%) who proceeded to HSCT developed SOS [26]. There were two additional smaller retrospective studies published by the French and Italian groups, which revealed similarly impressive CR rates and comparable toxicity profile in heavily pretreated patients (Table 1) [27,28].

The European Innovative Therapies for Children with Cancer Consortium’s prospective Phase I/II multicenter, single-arm, open-label study (ITCC-059) of InO monotherapy in children confirmed the FDA-approved BSA-based fractionated dosing of 1.8 mg/m^2^/course to be the RP2D in the first prospective therapeutic study in children. In the Phase I portion of this study, 25 children (23 evaluable for DLTs) ages 1–18 years with R/R CD22+ ALL were treated with single-agent InO. ORR after course 1 was 80%, 84% of whom achieved an MRD-negative CR. Nine patients went on to receive HSCT or CART, and the 12 month ORR was 40%. Toxicity was similar to that observed in adults, primarily consisting of Grade 1–3 hepatotoxicity and hematologic toxicity. Grade 3–4 SOS was observed in two patients during subsequent chemotherapy but was not observed during subsequent HSCT as opposed to the high rates reported previously [29]. The Phase II component of the study enrolled 32 patients (28 treated, 27 evaluable). ORR was 81.5%, of whom the majority (81.8%) of them achieved an MRD-negative CR. One-year EFS was 36.7% and OS was 55.1%. Eighteen patients proceeded to consolidation therapy with HSCT and/or CART. SOS occurred in 7 of these patients [30].

**Table 3 biomedicines-11-01886-t003:** Pediatric Clinical Trials with Inotuzumab.

Patient Population Number of Patients(n) Age Range	Study Phase NCT# Design	Outcome	Toxicity- SOS	Refs.
R/R 51 0–21 years	Retrospective	Of 42 with M2/M3 marrow: 67% CR, of whom 71% MRD-neg	6 (12%) Grade 3 hepatic No SOS during therapy 11/21 (52%) of HSCT patients—SOS	[26]
R/R 12 3–18 years	Retrospective	8/12 CR/Cri, 2 of whom MRD-neg 1-year EFS 33% & OS 38%	4 (33%) Grade 3/4 hepatic 2/4 HSCT patients—SOS	[27]
R/R 16 0.5–18 years	Retrospective	68.7% CR, all MRD-neg 1-year EFS 27.5% & OS 45.8%	2 Grade 3 SOS with HSCT	[28]
R/R Phase I: 25 Phase II: 32 1–18 years	I/II (ITC-059) Monotherapy	Phase II: 1-year EFS 36.7%/OS 55.1%	7/18 SOS in HSCT or CART	[30]
R/R Cohort 1: 48 1–21 years	II NCT02981628 (COG AALL1621) Cohort 1: Monotherapy Cohort 2: InO + chemo	Cohort 1: CR/CRi 58.3%, 66.7% of whom MRD-negative Cohort 2: ongoing	6/21 Grade 3 SOS in HSCT	[31]
Frontline	III NCT03959085 (COG AALL1732) Chemo +/− InO	Ongoing		

COG AALL1621 (NCT02981628) is an ongoing single-arm, open-label Phase II trial for children with CD22+ R/R ALL. In the completed cohort 1, 48 evaluable patients were treated with InO monotherapy. The rate of CR/CRi after 1 cycle was 58.3%, 66.7% of whom were MRD-negative. InO was well tolerated with the most common toxicity being myelosuppression and a high rate of SOS with subsequent HSCT (28.6%), but rates of febrile neutropenia and infection were much lower than reported in similar patients receiving chemotherapy [31]. The study is now evaluating cohort 2 with integration of InO into a modified augmented BFM chemotherapy backbone, which is ongoing.

In light of the impressive outcome results with InO in the R/R setting, COG has also moved InO forward to the frontline setting for children and adolescent/young adults with newly diagnosed NCI high risk B-ALL (COG AALL1732) (NCT03959085). In this ongoing study, which opened in 2019, patients are randomized to receive the SOC intensive chemotherapy backbone with or without the addition of two blocks of InO for post-consolidation treatment.

In summary, inotuzumab has demonstrated remarkable efficacy in the R/R setting with tolerable toxicity and has thus been carried forward into the frontline setting for high-risk patients. However, early experience with combination therapy raises concerns about serious toxicity risks. Studies are currently ongoing and in development to evaluate how to best combine InO and chemotherapy in a safe yet effective way.

#### 1.1.3. Chimeric Antigen Receptor T (CAR T)-Cell Therapy

The evolution of CAR T-cell therapy has made tremendous advances since the first CAR T trial for childhood R/R B-ALL. CARs are autologous T cells genetically manufactured to target specific antigens on leukemic blasts and couple them with intracellular T-cell signaling domains of the T-cell receptor (TCR), therefore re-directing T-lymphocytes to leukemic blasts expressing the targeted antigen. There are now five generations of CARs in development, all employing different strategies to enhance the signal transduction leading to stronger activation, expansion, and persistence of CAR T-cells while also limiting associated toxicities such as CRS and neurotoxicity [32,33,34]. The FDA approval of tisagenlecleucel (Kymriah) in 2017 for patients up to age 25 with refractory or second or greater relapse of B-ALL significantly impacted the landscape of relapsed ALL in children and young adults.

CD19 CAR T-Cell TherapyFDA Approved CD19 CAR T Therapy

There are currently four CD19 CAR T products that are FDA approved: tisagenlecleucel (Kymriah), which was approved in 2017 and is the only product to be approved for R/R pediatric B-ALL; axicabtagene ciloleucel (Yescarta), which was also approved in 2017 for R/R B-cell lymphoma in adults; brexucabtagene autoleucel (Tecartus), which was approved in 2020 for R/R B-ALL in adults and is the sole CAR T therapy approved for R/R mantle cell lymphoma; and lisocabtagene maraleucel (Breyanzi), which was approved in 2021 for R/R B-cell lymphoma in adults [35]. To date, CD19 CAR T-cell therapies have only been widely approved for B-ALL and chronic hematological diseases in adults. Other CAR T products targeting various antigens in both hematological malignancies and solid tumors are in different stages of research and clinical development.

Mechanism of Action

The first generation of CAR T cells used a single intracellular domain CD3z coupled with the extracellular domain responsible for antigen recognition, which resulted in a short duration of CAR-T durability and led to treatment failure. The second generation of CAR T cells employed an additional intracellular signaling domain of costimulatory receptors to CD3z, such as 4-1BB or CD28, resulting in better durability and expansion of CAR T cells. Antigen markers such as CD19 and CD22 on ALL blasts are the most common antigens recognized by CARs to be coupled to 4-1BB and CD28 [36]. Third-generation CAR T cells utilize two additional costimulatory domains. Fourth and fifth generation CAR-T therapies are significantly more diversified, employing various cytokines to remodel tumor microenvironment and break the resistance of the malignant cells, with CAR-T constructs that are universal, self-driving, armored, self-destructive, or conditional [33,34,37].

All current FDA approved CD19 CAR-T therapies are second generation. Tisagenlecleucel and lisocabtagene maraleucel are autologous T cells transduced with a lentiviral vector to express a CAR-containing CD3-zeta domain with a 4-1BB costimulatory domain, while axicabtagene ciloleucel and brexucabtagene use a CD28 costimulatory domain [33,35].

The CD19 B-cell marker is an excellent target since it is highly expressed on a majority of B-lymphoblastic blasts while remaining absent on other lineage and non-hematopoietic cells [37].

Clinical Data in Pediatric B-ALL (See Table 4 for Further Details on Specific Clinical Trial Results)

A Phase I/II single-institutional study of tisagenlecleucel in thirty children and young adults with CD19+ R/R B-ALL (NCT01626495) showed a high remission rate of 90% at 1 month post-infusion, even in those patients who were previously refractory to blinatumomab and post-HSCT relapse. In addition, the 6-month EFS and OS rates were 67% and 78%, respectively. Only three patients underwent HSCT [38].

**Table 4 biomedicines-11-01886-t004:** Selected CAR-T clinical trials in pediatric ALL.

Patient Population- Number of Patients (Age Range)	Study Phase NCT#	Outcome	HSCT Post CAR-T Therapy	CAR-T Persistence	CRS and ICANS	Refs.
**CD19 CAR T**						
30 (25 pediatric patients 5–22 yrs and 5 adult patients 26–60 yrs)	Phase I/IIa NCT01626495 and NCT01029366	CR: 90% (88% MRD-negative) 6 mo EFS and OS: 67% and 78%	3 patients underwent HSCT: remained in remission at 7–12 months after CAR T therapy	Persistence at 6 mo: 68%	CRS: 100% (any grade), 27% (severe requiring ICU care) ICANS: 43% (any grade)	[38]
75 (3–23 yrs)	Phase I/IIa ELIANA Trial NCT02435849	3 mo ORR: 81% (100% MRD-negative) 6 mo EFS and OS: 73% and 90% 12 mo EFS and OS: 50% and 76% 36-month RFS, EFS and OS: 52%, 44% and 63%	8 patients underwent HSCT (including 2 with early B-cell recovery and 2 with +MRD)- no relapse to date.	Median duration: 168 days	CRS: 77% (any grade), 46% (grade ≥ 3) ICANS: 40% (any grade), 13% (grade 3), no grade 4 events.	[39,40]
20 (5–25 yrs).	Phase I NCT01593696	CR: 70% for B-ALL (60% MRD-negative) 10 mo OS: 51.6% LFS (for all 12 MRD-negative ALL): 79% at 4.8 mo.	10 patients underwent HSCT—remained in MRD-negative remission at a median follow up of 10 months	No detectable CAR T cells after Day 68, although a majority of patients underwent HSCT which likely eliminated it	CRS: 76% (any grade), 28.6% (grade ≥ 3) ICANS: 28.6% (any grade)	[41]
43 (1–25 yrs)	Phase I/II PLAT-02 Trial NCT02028455	MRD-negative CR: 93% 12 mo EFS and OS: 50.8% and 69.5%	11 patients underwent HSCT: 2 developed CD19+ relapses	6.4 months (median duration of B-cell aplasia as a measure of CAR-T persistence)	CRS: 93% (any grade), 23% (severe) ICANS: 49% (any grade), 21% (severe)	[42]
74 (72 B-ALL, 2 B-LLy) age 1–29 yrs CAR-naïve cohort: 41 Retreatment cohort: 33	Phase I Humanized Anti-CD19 CAR T NCT02374333	CAR-naïve MRD-negative CR#: 100% 12 mo RFS and OS: 84% and 90% 24 mo RFS and OS: 74% and 88% Retreatment: CR 64% (86% MRD-negative) 12 mo RFS and OS: 74% and 76% 24 mo RFS and OS: 58% and 55%	CAR-naïve: 4 patients proceeded to HSCT Retreatment cohort: 1 patient proceeded to HSCT due to early B-cell recovery	6 mo cumulative incidence of loss of CAR-T persistence: 27% (CAR-naïve) and 48% (retreatment cohort)	CAR-naïve: CRS: 90% (any grade), 15% (grade ≥ 3) ICANS: 41% (any grade), 7% (grade ≥ 3) Retreatment cohort: CRS: 76% (any grade), 15% (grade ≥ 3) ICANS: 36% (all < grade 3)	[43]
50 (4.3–30.4 yrs)	Phase I NCT01593696	CR: 62% (90.3% MRD-negative) At median follow up at 4.8 yrs OS: 10.5 mo EFS: 3.1 mo	21/28 in MRD-negative CR underwent HSCT and 2 relapsed. 7/28 in MRD-negative CR who did not proceed to HSCT all relapsed	N/A	CRS: 70% (any grade), 18% (grade ≥ 3) ICANS: 20% (any grade), 8% (severe)	[44]
24 (3–20 yrs)	Phase I/II NCT02625480 ZUMA-4	CR: 67% (100% MRD-negative) RP2D cohort (5/6 underwent HSCT): CR: 67% Median DOR, OS, and DFS were not reached. 24 mo OS: 87.5% RFS censoring and without censoring for HSCT: 5.2 mo vs. 7.4 mo	16 underwent HSCT (67%)	CAR T not detectable after 3 months (but a majority of patients proceeded to HSCT at a median 2.3 mo after infusion)	CRS: 88% (any grade), 33% (grade ≥ 3) ICANS: 67% (any grade), 21% (grade ≥ 3)	[45]
255 (0.4–26.1 yrs)	Non-interventional, prospective study	CR: 85.5% (99.1% MRD-negative) 12 mo DOR, EFS and OS: 60.9%, 52.4% and 77.2%	34 responders (16.1% of all patients) underwent HSCT	N/A	CRS: 55% (any grade), 16.1% (grade ≥ 3) ICANS: 27% (any grade), 9% (grade ≥ 3)	[46]
**CD22 CAR-T**						
21 (7–30 yrs)	Phase 1 NCT02315612	CR: 57% in all pts (75% MRD-negative) CR: 73% in dose ≥ 1 × 10^6^ CD22-CAR/kg (9/12 MRD-negative)	N/A	Day 28: 15/21 CAR T- cells detectable in blood 3 mo: 7/9 CAR T-cells detectable in blood	CRS: 76% (all < grade 3) ICANS: 37.5% (all mild or transient)	[47]
56 (4.4–30.6 yrs)	Phase 1—Updated results of NCT02315612	CR: 72.7% for ALL patients (63.6% MRD-negative) Median OS and RFS for responders: 13.4 mo and 6 mo	13 underwent HSCT and 6 subsequently relapsed.	Median percentage of CAR-positive T cells at peak expansion between 14–21 days post-infusion: 77%	CRS: 86.2 (any grade), 10% (grade ≥ 3) ICANS: 32.8% (all < grade 3 except for one grade 4)	[48]
17 (3–28 yrs)	Phase I NCT02650414	CR: 77% (77% MRD-negative) Median RFS, EFS, and OS: 5.3 mo, 5.8 mo and 16.5 mo	5 (in MRD-negative CR) underwent HSCT- 1 subsequently had a CD22+ relapse while 4 remained in CR 6 (in MRD-negative CR) did not undergo HSCT- 5 subsequently relapsed and 1 remained in CR at 30 mo post-infusion	Persistence correlate with clinical response	CRS: 88% (all < grade 3) ICANS: 35% (all < grade 3) 1 patient out of retreatment cohort experienced grade 3 CRS and ICANS	[49]
8 (5 children, 3 adults, no specific age reported)	Phase 1 NCT02588456 (adult) NCT02650414 (pediatrics)	12 mo CR: 50%	N/A	N/A	CRS: 75% (any grade), 12.5% (grade 3 in adult) ICANS: 0%	[50]
**Dual (CD19 and CD22) targeted CAR T**						
14 (8 pediatric and 6 adult patients: 2–68 yrs)	Phase I NCT03233854 NCT03241940	CR: 92% OS: 92% at median 9.5 months from infusion	6 pediatric patients underwent HSCT and 1 died while in CR from complication related to HSCT No adult patient underwent HSCT	N/A	CRS: 75% (all < grade 3 in pediatrics) ICANS: 17% (all < grade 3 in pediatrics) 1 adult with high disease burden experienced Grade 4 CRS and ICANS	[51]
15 (4–16 yrs)	Phase I/II AMELIA trial NCT03289455	1 mo CR: 86% 12 mo OS and EFS: 60% and 32%	1 patient underwent HSCT	Longer median duration of detection of 344 days for patients who received 3 × 10^6^ cells/kg than other dosages Median time to last detection: 119 days	CRS:80% (all < grade 3) ICANS: 27% (all grade 1)	[52]
194 (≤20 yrs)	Phase II Chinese Clinical Trial Registry: ChiCTR2000032211	CR: 99% (100% MRD-negative) 12 mo EFS and OS: 73.5% and 87.7%	78 patients underwent HSCT 12 mo EFS (HSCT vs. no HSCT): 85% vs. 69.2% (*p* = 0.03)	Detection by RT-PCR found that CD19 CAR-T expansion occurred earlier and for longer duration than CD22 CAR T (measured up to 660 days post-infusion).	CRS: 88% (any grade), 28.4% (grade ≥ 3) ICANS: 20.9% (any grade), 4% (grade ≥ 3) 2 patients died following infusion due to CRS and neurotoxicity	[53]
12 (<31 yrs)	Phase I NCT03330691	CR: 91% (100% MRD-negative)	N/A	N/A	CRS: 45% (all grade 1) ICANS: 45% (all grade 1 except one self-limited grade 3 event)	[54]
20 (5.4–34.6 yrs)	Phase 1 NCT03448393	CR: 60% (for entire cohort) CR: 71.4% (CAR-naïve cohort) 6 mo and 12 mo RFS for pts in CR: 80.8% and 57.7%	N/A	N/A	CRS: 50% (any grade), 15% (grade ≥ 3) ICANS: 5% (grade 3)	[55]
**Universal CAR T**						
21 (7 children and 14 adults 0.8–16.4 yrs)	Phase I NCT0280442 NCT02746952	CR: 67% (71% MRD-negative) Median duration of response: 4.1 mo. 6 mo PFS and OS: 27% and 55%	10 out of 14 responders underwent HSCT 6 mo PFS and OS: 27% and 55%	N/A	CRS: 91% (any grade), 14% (grade ≥ 3) ICANS: 38% (all < grade 3)	[56]

Abbreviations: mo: month; CR: complete remission; DOR: duration of remission; CR# of B-ALL patients only (without LLy patients). Please note that percentage of MRD-negative patients are among those already in CR.

A follow up Phase II, international, multi-institutional study using tisagenlecleucel in 75 pediatric patients with R/R B-ALL (ELIANA; NCT02435849) confirmed these encouraging results. The study showed a CR rate of 81% at 3 months post-infusion, all of whom were MRD-negative [39]. At 6 and 12 months, the OS was 90% and 76%, while the EFS was 73% and 50%, respectively. Since only eight patients underwent HSCT following CAR-T therapy, the outcomes were reflective of a cohort who largely did not receive further therapy after CAR-T therapy. Based on the results of this pivotal study, tisagenlecleucel was FDA approved in 2017 for patients ≤ 25 years of age with refractory or second or greater relapse of B-ALL, marking a tremendous achievement in the management of relapsed disease.

In longer-term follow-up analysis of 79 pediatric and young adult patients with R/R B-ALL, the ELIANA trial reported a 36-month RFS, EFS, and OS of 52%, 44%, and 63%, respectively [40], suggestive of similar durability at this timepoint. The relapse free survival (RFS) with and without censoring for interim therapy (including HSCT) was 52% and 47.8%, respectively at 36 months. Twenty-two percent of patients underwent HSCT, none of whom relapsed.

Other early studies of CD19 CAR-T therapy in children and young adults with R/R B-ALL have shown similar CR rates of 70–96% and attainment of MRD-negative remission in a majority of responders (60–93%), as well as similar EFS and OS [41,42].

Given its promising outcomes in pediatric R/R B-ALL, tisagenlecleucel is also being tested as upfront therapy in COG study AALL1721 (NCT03876769) for newly diagnosed, high-risk B-ALL patients who remain MRD-positive following consolidation phase. St Jude’s Total Therapy XVII protocol (NCT03117751) also incorporates 19-BBz (4-1BB with CD3) CAR-T therapy for patients with B-ALL plus either MRD ≥ 1% at the end of induction or with isolated CNS relapse. Because the intensification of frontline conventional chemotherapy has likely reached its threshold [25], the addition of tisagenlecleucel allows for intensification of anti-leukemic treatment without escalating toxicities related to conventional chemotherapy.

Another CD19 CAR-T product, brexucabtagene autoleucel (KTE-X19), was tested in a multicenter, phase I trial in 24 children and adolescents with R/R B-ALL (ZUMA-4; NCT02625480). The study showed an overall response rate (ORR) of 67% at 28 days post-infusion, all of whom were MRD-negative. In contrast to the ELIANA trial, a majority of the responders (88%) proceeded to HSCT and achieved a 24-month OS rate of 87.5%. For the 14 patients (58%) who were still alive, all of them had undergone HSCT following KTE-X19 infusion [45]. This suggests durability of KTE-X19, at least in those patients who achieved remission following CAR T-cell infusion followed by HSCT. Currently a Phase II trial is in process, and eligibility was broadened to include patients with MRD-positive disease and early relapses, defined as ≤18 months from diagnosis.

Registry data taken from the Center for International Blood and Marrow Transplant Research (CIBMTR) on the utilization of CD19 CAR-T therapy for R/R pediatric B-ALL revealed outcomes that are comparable to those of clinical trials, with a CR rate of 85.5%, 99.1% of whom were MRD-negative. The 12-month EFS and OS were 52.4% and 77.2%, respectively. Similar to the ELIANA trial, only 16.1% of patients underwent HSCT following CAR-T therapy [46]. The real-world patient population from the registry was more diversified than those in the ELIANA trial but still showed a similar outcome and safety profile. This suggests that a broader patient population than those studied in clinical trials may also benefit from tisagenlecleucel.

Challenges of CAR-T Therapy

It is important to note that while CAR-T therapy has significantly changed the landscape for the management of relapsed ALL in children, a significant proportion of children with R/R B-ALL may never receive CAR T cells due to limitations, which include manufacturing failures, feasibility, cost, toxicity, and/or death from progressive disease while awaiting the manufacturing or infusion of CAR T cells. In the ELIANA trial, 107 patients were screened, 92 were enrolled, but only 75 underwent infusion, which left 32 (30% of screened patients) patients who did not receive CAR-T infusion for a number of reasons [39]. Similarly, in the ZUMA-4 trial, 7 out of the 31 (23%) enrolled patients did not receive KTE-X19 CAR-T product [45]. Even for patients who are infused with CAR T cells, approximately 50% of patients experience relapse within 1 year of CAR-T therapy. Relapses occur through either loss of functional CAR T-cell persistence resulting in a CD19-positive relapse, escape from CAR T cells due to antigen loss resulting in a CD19-negative relapse, or more rarely through a lineage switch, especially in infant B-ALL with KMT2A-rearrangement [57]. It also remains unanswered whether CD19 CAR-T therapy should be used as monotherapy or as a bridge to HSCT.

Most CAR-T products that have been approved or are under clinical investigation contain single-chain variable fragment domains derived from mouse monoclonal antibodies that could lead to rejection from antimurine immune responses. In an effort improve persistence of CAR T cells, humanized CAR T (huCART) cells have been developed in the hope of bypassing the immunogenicity against murine domain and improving survival. A single-institutional pilot study of huCART19 in 72 pediatric patients with R/R B-ALL (NCT02374333), including 39 CAR-naïve patients and 33 retreated patients due to early B-cell recovery or non-responders with previous exposure to CAR-T therapy, showed a CR rate of 100% in the CAR-naïve cohort and a CR rate of 64% in the retreatment cohort, 86% of whom were MRD-negative. RFS for the CAR-naïve cohort at 12 and 24 months were 84% and 74%, respectively, one of the highest RFSs for this cohort of patients. RFS for the retreatment cohort at 12 and 24 months were 74% and 58%, respectively, suggesting that huCART19 can still achieve long-term persistence even in patients with a history of poor persistence of murine-derived CAR T cells [43].

The durability of remission following CAR-T therapy from the various CAR-T products that exist also remains unclear, as well as the role for HSCT while in CR following CAR-T therapy. The advantage of avoiding the toxicities related to HSCT must be weighed against the risk of relapse post-CAR-T therapy, for which outcomes are poor and limited by therapeutic options.

The ELIANA trial, which started with a high ORR 81%, showed a 6-month RFS of 80%, a 12-month RFS of 59%, and a 36-month RFS of 52%, suggesting that the initially high remission rate was not durable over time and that there is still room for improvement [39,40]. Other studies have also reported that approximately 50% of children and young adults may relapse within the first year following CD19 CAR-T therapy [42,58]. Therefore, further consolidative strategies following CAR T therapy may still be needed in subset of patients to optimize the durability of remission.

The longest follow-up to date for CD19 CAR-T therapy for R/R pediatric B-ALL was reported by Shah and colleagues [44]. In this cohort of 50 patients, 90.3% of responders were MRD-negative. Unlike the ELIANA trial, a majority of patients (75%) in a MRD-negative CR following CAR-T therapy proceeded to HSCT. At a median follow-up of 4.8 years, patients who underwent HSCT had a low relapse rate of only 9.5% at 24 months post-CAR-T, a 5 yr EFS of 61.9%, and a median OS of 70.2 months post-transplant, while patients who did not undergo HSCT all relapsed. This is in contrast to the Park et al. study, which showed HSCT for adult patients in MRD-negative CR following CD19 CAR-T therapy did not impact EFS or OS [59], although this may be attributed to an adult-age cohort who likely had a high mortality and morbidity related to HSCT compared to the pediatric population.

Although the Shah et al. study showed a clear benefit for HSCT with a high relapse rate for those who did not proceed to HSCT, the role of HSCT following CAR-T therapy remains uncertain. Only a minority of responders (13–16% of patients) proceeded to HSCT on the ELIANA trial and in real-world data [39,46], in contrast to the higher proportion of responders who proceeded to HSCT as reported by Shah et al. and the ZUMA-4 study (67–75%). A prospective trial incorporating HSCT following CAR-T therapy is needed to better distinguish the cohort of patients who may benefit from HSCT following CAR-T therapy. Better identification of the pre-infusion factors predisposed for relapse following CAR-T therapy would also help inform the role of HSCT. For now, the decision of HSCT is very much up to the physician’s judgement and the family’s wishes.

Adverse Events Related to CAR T-Cell Therapy

Adverse events following CAR-T therapy primarily include CRS and immune effector-cell-associated neurotoxicity syndrome (ICANS). B-cell aplasia leading to hypogammaglobulinemia and the need for IVIG replacement has also evolved as not only a common adverse effect but also an indicator of CAR-T persistence and durability.


*
Cytokine Release Syndrome (CRS)
*


CRS is a well-recognized adverse effect of CAR-T therapy due to the expansion of CAR T cells and tumor cell apoptosis followed by an inflammatory response consisting of supraphysiological levels of interluekin-6, tumor necrosis factor (TNF), and activated T cells. CRS can range from mild and self-limited with fever only to severe cases with hypotension, hypoxia, respiratory failure and capillary leak syndrome. It has an onset between 1 to 7 days post-infusion (median onset 3 to 5 days) and a median duration of 5 to 8 days [38,39,43,45]. Although there are many grading systems for CRS, COG studies which have incorporated CD19-directed immunotherapy (AALL1731/NCT03914625 and AALL1721/NCT03876769) have used the Lee Criteria for CRS grading [41]. CRS of any grade is common among patients receiving CD19 CAR-T therapy with a reported incidence of 70–100% across all clinical trials, while ≥Grade 3 CRS is less common and reported in 15–46% of all studies. In real-world experiences with tisagenlecleucel, CRS of any grade was reported in 55% of patients, while ≥Grade 3 CRS was reported in 16.1% of patients [46], suggesting a higher safety profile than in clinical trials. It is also noteworthy that while the ELIANA trial had a median time of 45 days from enrollment to infusion [39], the ZUMA-4 trial had a median time of only 16 days from leukapheresis to product release [45], which likely resulted in a higher CRS rate.

Interventions for CRS include immunosuppressives such as corticosteroids, anti-IL-6 therapy tocilizumab, and general supportive care measures in intensive care units such as vasopressor, fluid, supplemental oxygen, and ventilatory support. CRS is usually fully reversible, but close monitoring is necessary to initiate early timing of interventions. Immunosuppressive measures for CRS have been shown to decrease progression to more severe CRS and did not negatively impact the anti-leukemic efficacy of CD19 CAR-T therapy or disease-free survival [60,61].


*
Immune effector-cell-associated neurotoxicity syndrome (ICANS)
*


ICANS is a spectrum of diverse neurotoxicity that can follow CAR-T therapy. Symptoms range from mild, such as subtle confusion or agitation, to severe, such as encephalopathy, lethargy, aphasia, difficulty focusing, tremors, or seizures. Most symptoms are transient and fully reversible if recognized early. ICANS has a median onset of 6 days post-infusion and a median duration of 6 days. It can occur with or without CRS following CAR T therapy [62]. Both the ELIANA and ZUMA-4 trials reported that neurological events occurred mainly during CRS or shortly after its resolution [39,45]. Multiple studies have associated neurotoxicity with high-grade CRS [39,44]. The pathophysiology of ICANS is poorly understood, but studies have shown that increased permeability of the blood–brain barrier leads to a high level of cytokines, which may result in the development of overt neurotoxicity in patients with ICANS [63]. ICANS of any grade is reported in 20–67% of patients treated with CD19 CAR-T therapy, although ≥Grade 3 ICANS is only reported in 7–21% of patients. Prompt recognition and interventions are necessary to avoid a rapid deterioration in neurological status. Optimal management include dexamethasone for CSF penetration or high-dose methylprednisolone pulse with the addition of tocilizumab if there is concurrent CRS present [61,62].


*
Hypogammaglobulinemia
*


CD19 CAR T-cell-mediated B-cell aplasia resulting in hypogammaglobuinemia is frequently seen following CD19 CAR-T therapy. This requires routine replacement of intravenous immunoglobulins (IVIGs) to maintain IgG level above 4–5 g/L to prevent infections. Continued B-cell aplasia and resulting hypogammaglobuinemia is related to persistence of CD19 CAR as all responders in the ELIANA trial had B-cell aplasia and received IVIG [39]. Long-term follow-up of the ELIANA trial demonstrated that B-cell aplasia persisted in 71% and 59% of responders at 12 and 24 months while also demonstrating that patients who lost B-cell aplasia in <6 months post-infusion had a much shorter duration of remission [40]. This is in line with the Pulshipher et al. study, which also showed that loss of B-cell aplasia at <6 months from infusion was highly predictive of relapse.

#### 1.1.4. CD22 CAR T-Cell Therapy

CD19-negative relapses following CD19 CAR-T therapy is a well-known failure. A recent multi-institutional retrospective review of the phenotypic pattern of relapse following CD19 CAR-T therapy revealed that 41.7% of relapses were associated with the loss of CD19 antigen [57]. Treatment options are very limited for these patients with a median OS of only 9.7 months following CD19-negative relapse [57]. In these cases, the expression of CD22, which is also expressed by a majority of B lymphoblasts, is usually retained on ALL blasts and may be an alternative immunotherapeutic target.

FDA Approved CD22-Directed CAR T: NoneMechanism of Action

CD22-directed CAR T-cell therapy employs similar strategies as CD19 CAR-T therapy. Most CD22 CAR-T products under investigation also use a 4-1BB or CD28 costimulatory domain.

Although CD22 expression is present on the majority of B-ALL blasts at diagnosis and is usually retained in patients with CD19-negative relapse following CD19 CAR-T therapy, its expression is more variable than CD19 expression, which potentially could select for more pre-existing CD22-negative or CD22-low-expression blasts following CD22-directed treatment, resulting in a higher risk of relapse.

Clinical Data in Pediatric B-ALL (See Table 4 for Further Details on Specific Clinical Trial Results)

The first CD22 CAR-T therapy in pediatric B-ALL, conducted at National Cancer Institute (NCI), was a Phase I trial using a 4-1BB costimulatory domain (NCT02315612). The initial report consisted of 21 children and young adults with R/R CD22+ B-ALL, including multiply relapsed patients who had all undergone HSCT and 15 of whom had also received prior CD19 CAR-T therapy. The report revealed a CR rate of 73% in those patients receiving ≥1 × 10^6^ CD22-CAR T cells/kg—with 9 out of 12 responders also achieving MRD-negative remission. Eight out of the twelve responders relapsed at a median of six months post-infusion, which was associated with diminished CD22 expression in seven out of eight relapses. Rather than complete loss of targeted antigen as in CD19 CAR T, the pattern of CD22 expression in relapses is more variable and diminished rather than complete loss. In vivo studies demonstrated that diminished CD22 expression is a mechanism for relapse following CD22 CAR-T therapy, suggesting that escape from CD22 CAR T-cells is likely possible [47]. Updated study results in 2020 including a total of 58 patients continued to show a high CR rate of 72.7%, 63.6% of whom were MRD-negative [48]. The median OS was 13.4 months and the RFS was 6 months. Thirty out of the forty patients in CR (75%) subsequently relapsed, the majority of whom were CD22-negative or dim disease. Although the numbers were small, patients who underwent HSCT had more favorable outcomes related to RFS and EFS. Notably, prior response to CD19 CAR-T therapy or HSCT did not impact response to CD22 CAR-T therapy.

A smaller study carried out by Singh and colleagues at CHOP and University of Pennsylvania utilizing a similar CD22/4-1BB-based CAR as the NCI study included five children and three adults who had previously all undergone CD19-directed therapy and most of whom had CD19-negative leukemia (NCT02588456 and NCT02650414) [50]. The study unexpectedly showed poor outcomes, with only a 50% CR rate. Although there were some transient responses, overall outcomes were clearly inferior to the NCI study. Further preclinical work showed that a small alteration in the structure of CD22 CAR T, namely the shorter length of the scFv linker in the CAR construct used in the NCI study, can enhance anti-leukemic efficacy of CAR T-cells.

A follow-up single-institutional Phase I study of a modified CD22/4-1BB CAR construct, utilizing a shorter scFv linker, in 17 children with CD19-negative relapse of B-ALL following CD19-directed therapy showed a high CR rate of 77%, 77% of whom were MRD-negative (NCT02650414). The median RFS, EFS, and OS were 5.3 months, 5.8 months, and 16.5 months, respectively at a median follow-up of 29 months. The safety profile was also favorable [49].

Adverse Events Related to CD22 CAR T-Cell Therapy

CD22 CAR studies reported an incidence of 75–88% of CRS and 0–37% of ICANS, with only a few cases being at least Grade 3. Similar to CD19 CAR T, CRS had a median onset of 5–7 days after infusion and a median duration of 5 days. Neurotoxicity associated with CD22 CAR-T therapy was more frequently milder grade and transient than neurotoxicity associated with CD19 CAR-T therapy [47,48]. There are, however, some distinct adverse effects between the two immunotherapies. Some of the unexpected distinctions reported by Shah and colleagues included 5% incidence of atypical hemolytic uremic syndrome, 5% incidence of severe capillary leak syndrome out of proportion to CRS, and 21% incidence of ocular symptoms following CD22-directed CAR-T therapy. In addition, hemophagocytosis lymphohistiocytosis (HLH)/macrophage activation syndrome (MAS) occurred in 38% patients with CRS (and only in patients who experienced CRS), necessitating treatment with anakinra and steroids in some patients. The onset of HLH/MAS was typically after CRS was resolving or had already resolved, suggesting a unique pathophysiology distinct from CRS [48]. Myers et al. also reported some uncommon adverse effects including platelet refractoriness and inflammatory reaction to platelet transfusion in one patient and delayed HLH in another patient following CD22 CAR-T therapy [49].

Overall, CD22 CAR-T trials showed a similar outcome with CR rate of 73–77% and safety profile compared to results from CD19 CAR-T trials, suggesting a viable alternative treatment option in this highly refractory population.

#### 1.1.5. Dual Targeting (CD19/CD22) CAR T-Cell Therapy

In addition to CD19-negative relapses following CD19 CAR-T therapy, downregulation of CD22 antigen has also been observed following CD22 CAR-T therapy [47,48]. Based on previous evidence for CD19-antigen loss or downregulation following CD19 CAR-T therapy resulting in CD19-negative relapse, it is likely that downregulation or loss of CD22 antigen also contributed to resulting relapse.

With the rationale that downregulation of both CD19 and CD22 antigens simultaneously on a single leukemic blast is unlikely, CAR T-cell therapy with dual targeting of CD19 and CD22 have been developed in children and young adults with R/R B-cell ALL to overcome antigen escape by CD19 or CD22 single-antigen targeting.

##### Mechanism of Action [51,52,53,54,64]

Dual targeting of CD19 and CD22 can be achieved in three different ways:

1. Co-transduction of T cells with two vectors encoding two separate CARs

2. Bicistronic CAR-T therapy in which two individual CARs exist in the same transduced T cells

3. Simultaneous, co-administration of two separate CAR-T products (CD19 and CD22 CAR T cells).

##### Clinical Data in Pediatric ALL (See Table 4 for Further Details on Specific Clinical Trial Results)

AUTO3 is a dual-targeted (against CD19 and CD22), bicistronic CAR T-cell therapy, incorporating tumor necrosis factor (TNF) as a co-stimulatory domain. It was studied in a Phase I trial (NCT03289455) including 15 children and young adults with R/R B-ALL, most of whom were CAR-naïve (93%). The study showed a CR rate of 86% (13/15) at 1 month post-infusion. Of the 13 patients in CR, 9 ultimately relapsed, 1 remained in CR without further treatment, and 3 received other ALL-directed treatment while in CR, including one who underwent HSCT. Eight of the nine relapses occurred with low CAR T cells, while 5 had detectable B-cells, indicating that insufficient long-term CAR-T persistence was the main problem. Persistence of CAR T-cells was detected until a median of 119 days compared to the ELIANA study, which last detected CAR T cells until a median of 168 days. The 1-year OS and EFS were 60% and 32%, which are inferior to the ELIANA trial, suggesting that better strategies are needed to increase the durability of this dual-targeted CAR-T construct [52]. The study did not show any increased toxicity compared to either CD19 or CD22 CAR-T therapy alone, without any reported Grade 3 neurotoxicity or CRS, demonstrating a favorable safety profile.

The method of simultaneous co-administration of CD19 and CD22 CAR T cells is another strategy of dual antigen targeting. The safety and efficacy of this method was demonstrated through a large Phase II multicenter trial conducted in China consisting of 194 pediatric patients with bone marrow involvement of R/R B-ALL [53]. The study achieved an impressive CR rate of 99% (N = 192) at 28 days post-infusion, all of whom were MRD-negative. The 12-month OS and EFS were 87.7% and 73.5%. Relapses occurred in 43 patients (22%) at a median follow-up of 11 months post-infusion, resulting in 1 yr PFS of 52.9%. Although the results cannot be directly compared to the ELIANA trial or real-world registry data, it does suggest improved CR rates and more durable remission. Long-term outcomes will reveal the durability of remission.

The majority of the relapses occurred as CD19+/CD22+ without loss of antigen, suggesting that there was limited persistence of CAR T cells. Persistence of CAR T cells for both CD19 and CD22 were short with a median time to B-cell recovery of 4 months. All 11 relapses that were phenotypically CD19+/CD22+ had lost CD19 and CD22 CAR T-cell persistence at the time of relapse.

Although the majority of patients did not receive further therapy following CAR T, the 12-month EFS was significantly higher for the 78 patients who underwent HSCT while in CR following CAR-T therapy, which included patients who were at risk for myeloid lineage switch due to underlying KMT2A-rearrangement or ZNF384-rearrangement or parental preference, compared to those who did not undergo HSCT (85% versus 69.2% respectively; *p* = 0.03). This again supports emerging data that HSCT is likely needed in a subset of high-risk patients.

This study was remarkable in that it is the largest prospective CAR-T trial for pediatric ALL to date with an impressively high CR rate. The study was also novel in using peripheral blood to manufacture the CAR T-cell products, rather than leukapheresis, which led to faster turnaround time of about 1 week for the manufacturing of CAR T cells. This also contributed to less potential for disease progression while awaiting CAR-T manufacturing and higher CR rate following CAR-T infusion.

There are also three actively enrolling early phase trials utilizing dual antigen targeted CAR-T therapies.

Schultz et al. conducted a Phase I trial consisting of 19 pediatric and adult patients with R/R B-ALL utilizing a bivalent CAR construct with dual targeting of CD19/CD22 (NCT03233854 and NCT03241940). For the 12 patients in their interim data analysis, the CR rate was 92% at 28 days post-infusion. There were three relapses to date, all CD19-positive at the time of relapse due to short product persistence. All six pediatric patients in CR following CAR T therapy underwent HSCT, and one patient died in CR from treatment related complication. The OS for all infused patients was 92%. Nearly all cases of CRS and ICANS were lower than Grade 3; only one adult patient experienced Grade 4 CRS and ICANS due to high disease burden at the time of infusion [51].

PLAT-05 is a Phase I multicenter trial studying SCRI-CAR19x22v2, a dual-transduced CAR-T product with lentiviral vectors encoding for either a CD19- or CD22-specific CAR with 4-1BB co-stimulation, re-engineered from version 1, which was predominated by CD19 CAR population rather than CD19+/CD22+ population (NCT03330691). Twelve pediatric patients with R/R B-ALL received the infusion, most of whom had prior exposure to CD19 or CD22 CAR-T therapy. The study showed a CR rate of 91%, all of whom were MRD-negative [54]. Further work is ongoing to better optimize the dual targeting of CD19 and CD22 and to ensure that there is durability, balance, and persistence of both CD19 and CD22 constructs.

A Phase I study of a novel murine stem cell virus CD19/CD22-4-1BB bivalent CAR T-cell product showed a CR rate of 60% among twenty children and young adults with R/R B-ALL (NCT03448393). The CR rate was higher at 71.4% in CAR-naïve patients. The 6- and 12-month RFS were 80.8% and 57.7%, respectively. It was also well tolerated with mainly low-grade CRS and ICANS [55].

##### Adverse Events Related to Dual Targeting (CD19/CD22) CAR T-Cell Therapy

Studies on dual targeted CD19/CD22 CAR T reported an incidence of 45–88% of CRS and 17–45% of ICANS, which is comparable to that for CD19 or CD22 CAR-T therapies. There was less frequency of ≥grade 3 CRS and ICANS compared to CD19 CAR T. The study by Wang and colleagues is unique in their fast turnaround time of 1 week for manufacturing of CAR T cells, allowing for infusion of fresh, rather than cryopreserved, CAR T cells. Robust and rapid expansion of fresh CAR T cells likely resulted in an earlier median onset of CRS of only 1 day, more cases of ≥Grade 3 CRS (28.4%) and ICANS (4%), as well as high frequencies of Grade 3–4 hypotension (41.3%) and seizures (14.2)—especially among patients with CNS leukemia [53].

Other pre-clinical work that has been under development includes CAR-T products targeting three antigens in B-ALL (CD19, CD20, and CD22), which have shown anti-leukemic efficacy in cell lines, primary patient samples, and animal models [65].

#### 1.1.6. Universal CAR T-Cell Therapy

Another exciting development in CAR T-cell therapy is the development of a universal CAR T (UCAR T). UCART comes from an unmatched donor and uses gene-editing technologies such as zinc-finger nuclease (ZFN), transcription-activator-like effector nuclease (TALEN), and CRISPR-Cas9 [66,67].

This “off-the-shelf” approach would overcome the limitation that current autologous CAR-T products are individualized and manufactured based on the specific patient, making it not readily available due to a time-consuming manufacturing process. It typically takes 6 to 8 weeks to manufacture a patient-specific CAR-T product, time that not all R/R patients can afford. UCART is potentially less expensive, allows for faster treatment with a reduced need for bridging chemotherapy while awaiting CAR-T products, and broadens access by not requiring individually tailored CAR T cells [68]. The use of UCART would also offer a treatment option for those patients who develop profound lymphopenia or impaired function of lymphocytes following cytotoxic chemotherapy or HSCT, limiting a successful autologous harvest.

The first demonstration of the utility and safety of UCART occurred in two infants with R/R B-ALL, who achieved CR following CD19 CAR T cells using TALEN gene-editing technology to disrupt the gene encoding for T-cell receptors (TCR) and the gene encoding for CD52 [69]. These disturbances helped to minimize the risk of graft-versus-host disease (GVHD) and to acquire resistance to anti-CD52 alemtuzumab so that the CAR T cells are resistant to destruction and not eliminated from recipients when receiving alemtuzumab as a conditioning agent [32,34,66,68]. Both infants were able to proceed with HSCT following CAR T successfully.

Two early Phase I studies on UCART (NCT0280442 and NCT02746952), which enrolled 7 children and 14 adults with R/R B-cell ALL, reported a CR rate of 67% at 28 days post-infusion, 6-month PFS and OS of 27% and 55% and allowed for 10 out of the 14 responders (71%) to proceed with HSCT [56]. CRS was the most common adverse effect, seen in 91% of patients, but only 14% were ≥grade 3. There was also one case of grade 1 acute skin GVHD, a problem that would not occur with autologous CAR T. The study demonstrated anti-leukemic efficacy of UCART with a tolerable safety profile in this heavily pre-treated high-risk population.

Although conventional chemotherapy followed by HSCT is regarded as the “gold standard” for the management of R/R pediatric B-ALL, complications related to HSCT and continued poor outcomes even after HSCT, especially in certain high-risk molecular or cytogenetic populations, signifies the need for novel strategies to improve efficacy and reduce toxicity. Immunotherapy-based approaches such as CAR-T technology is emerging as a promising option in the management of R/R B-ALL.

### 1.2. T-ALL

T-ALL is a rare sub-type of ALL in children, representing about 15% of all cases. About 15–20% of pediatric patients with T-ALL will be refractory or relapse and face dismal outcomes with a 3-year EFS of <15% [70,71]. Due to a high rate of chemo-refractoriness in this population, novel therapy approaches are critically needed.

Daratumomab is a human immunoglobulin G1kappa monoclonal antibody that binds CD38 (Figure 1). It is FDA-approved for adults with relapsed multiple myeloma in combination with chemotherapy [72] and has been identified as an attractive agent for relapsed T-ALL due to robust CD38 surface expression on T-ALL lymphoblasts. CD38 is expressed at very low levels on normal hematopoietic cells, rendering it an ideal target. Daratumomab has shown promising efficacy in 14 of 15 patient-derived xenograft models of relapsed T-ALL [70,71].

In the Phase II open-label DELPHINUS study, children and young adults (age 1–30 years) with relapsed/refractory T-ALL or T-lymphoblastic lymphoma (T-LL) received combination daratumomab + standard cytotoxic chemotherapy for two cycles followed by HSCT. Twenty-nine T-ALL patients and ten T-LL patients received at least one dose of daratumomab. The ORR (CR + CRi) in pediatric patients was 83.3%, 60% in adolescent/young adult (AYA) ALL patients, and 40% in T-LL patients. Among pediatric ALL patients, 10 (41.7%) achieved MRD-negative CR. All pediatric patients experienced a grade 3–4 AE, but no patient discontinued daratumomab due to an AE [73,74].

Based on these promising results, a new COG upfront T-ALL/T-LL clinical trial is currently in development with a plan to incorporate randomization to daratumomab in addition to cytotoxic chemotherapy for high risk T-ALL. 

Unfortunately, CAR-T therapy is more challenging for T-ALL than for B-ALL because there are no targeted antigens exclusively for malignant T cells that do not overlap with normal T cells. Administration of CAR T cells against a shared antigen can cause T-cell aplasia and inhibit the expansion of CAR T cells [75]. Based editing has emerged as a method of precise, targeted single-nucleotide changes in the genome without DNA breaks rendering affected genes inactive. CAR-T therapy against T-cell malignancies is being re-engineered so that pan-T-cell markers such as CD3 and CD7 are knocked out or knocked down through genome based-editing technology so that the re-engineered T cells can target malignant T cells without fratricide [76,77]. Development of base-editing CAR-T therapy has shown efficacy in pre-clinical studies involving cell lines, patient samples, and patient-derived xenografts of T-ALL. For instance, Diorio and colleagues reported on the development and anti-leukemic activity of a quadruple-base-edited CAR T with simultaneous disruption of four genes, including the CD7 gene to avoid fratricide after transduction with a CD7-specific CAR, a TRAC gene to minimize graft versus host disease, a PD1 (programmed cell death 1) gene to improve antitumor performance, and a CD52 gene to enable conditioning with an anti-CD52 monoclonal antibody [78].

The world’s first therapeutic use of base-edited CAR T-cells for children with relapsed T-ALL was carried out by Chiesa and colleagues, who generated a base-edited CAR7 (BE-CAR7) with inactivation of three genes encoding CD52 and CD7 receptors and the β chain of the αβ T-cell receptor to evade lymphodepleting serotherapy, fratricidine, and GVHD, respectively. The interim results of their phase 1 trial (ISRCTN15323014) consisted of three patients. The first patient enrolled was a 13-year-old girl with relapsed T-ALL after HSCT, who achieved molecular remission within 28 days after infusion of BE-CAR7, allowing her to proceed with a nonmyeloablative HSCT with continued remission in the bone marrow at 9 months after HSCT. The same BE-CAR7 also showed anti-leukemic activity in the other two patients, allowing one patient to proceed to HSCT, while the other patient developed a fatal fungal complication [79]. Based on these promising pre-clinical and early clinical data on base-editing CAR T, this technology may ultimately expand the treat options for children with relapsed and refractory T-ALL.

## 2. Conclusions

In summary, the novel immunotherapies that have been established as well as those that remain under development for pediatric ALL represent promising anti-leukemic treatments that can lead to dramatic improvements in outcome for R/R ALL as well as reduce toxicity related to conventional chemotherapy. Currently, blinatumomab has FDA indications in pediatric ALL in first or second CR with MRD ≥0.1%. Given its promising results in R/R ALL populations, it is currently being tested in upfront therapy through COG AALL1731 in combination with standard cytotoxic chemotherapy in patients with NCI SR B-ALL with certain risk factors for relapse. In addition, InO and daratumumab have also shown promising efficacy in R/R B-ALL and T-ALL, respectively, in children and young adults. InO is currently being investigated in combination with chemotherapy for R/R pediatric ALL, as well as in upfront therapy for NCI HR B-ALL patients for post-consolidation treatment through COG AALL1732. Daratumumab is planned to be investigated as upfront therapy for high-risk T-ALL in the next COG clinical trial for de novo T-ALL. Furthermore, the FDA approval of tisagenlecleucel, a typical representation of CAR T-cell therapy, has significantly impacted the landscape of relapsed ALL in pediatrics and provided us with a promising therapeutic option when few others were available. Still, even outcomes following CAR-T therapy have revealed that approximately 50% of patients may relapse within 1 year of infusion. There is clearly still room for improvement as well as optimization of how and when we can utilize these therapies to maximize their benefits. The continued integration and expansion of these novel therapies as both upfront therapy and salvage regimens will further improve efficacy and reduce toxicity.

## Figures and Tables

**Figure 1 biomedicines-11-01886-f001:**
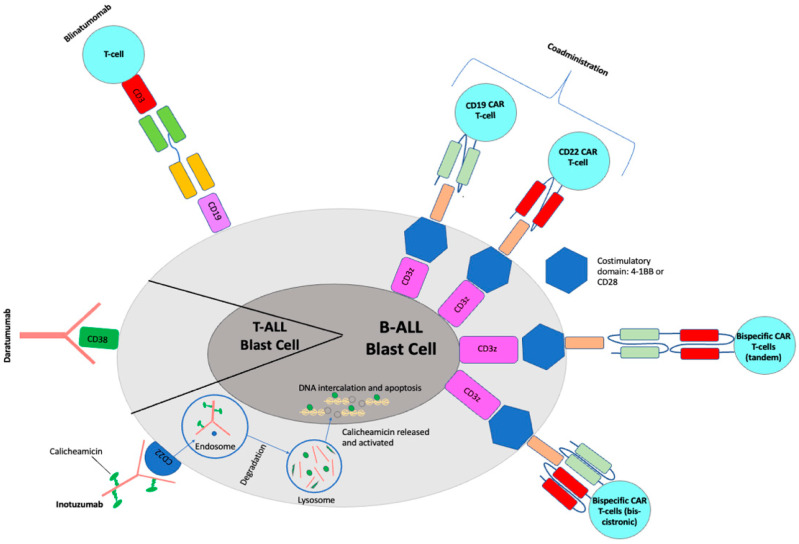
Mechanisms for current immunotherapeutic strategies for pediatric ALL.

**Table 2 biomedicines-11-01886-t002:** Risk Stratification for COG AALL1131 (NCT02101853).

High Risk	Early (<36 months) marrow Early (<18 months) isolated extramedullary (IEM)
Intermediate Risk	Late (≥36 months) marrow, End-Block 1 MRD ≥ 0.1% Late (≥18 months) IEM, End-Block 1 MRD ≥ 0.1%
Low Risk	Late (≥36 months) marrow, End-Block 1 MRD < 0.1% Late (≥18 months) IEM, End-Block 1 MRD < 0.1%

## Data Availability

Not applicable.

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
