# Peer review of "Harnessing the Immune System: Current and Emerging Immunotherapy Strategies for Pediatric Acute Lymphoblastic Leukemia"

_biomedicines, 2023, doi:10.3390/biomedicines11071886_

Round 1
Reviewer 1 Report
The manuscript offers a broad overview on immunotherapy in pediatric ALL. I have only minor comments.
- Chapter 1.1.1: CRh, incomplete hematologic recovery should be defined; BLAST study: it should be specified that RFS is at 18 months
- Table 1, at the bottom: Downs shold be Down
- Chpater 1.1.2: at the beginning: the study reported (first in Table 1 and reference 14) should be defined as in the paper as “conducted in 26 European and US Centers”. The I-BFM is a consortium not restricted to Europe. Reference 15 at the end of this paragraph is not appropriate. Data published in 2016 (reference 14 could not be compared with those pblished in 2021 (reference 15).
- Chpater 1.1.2: the study in reference 15 is not a BFM and AIEOP study. It can be defined as an IntReALL study conducted in the frame of the I-BFM Study Group consortium. Please note that Munster should be Münster. In the same paragraph “3rd concolidation” should be “3rd consolidation course”.
- Chpater 1.1.2: at the end it can be added also that the “AIEOP-BFM ALL 2017 study, is conducting randomized studies in patients at intermediate and high risk EUDRACT Number 2016-001935-12; ClinicalTrials.gov Identifier: NCT03643276: see below
- Randomization R-HR: High-risk (HR) pB-ALL defined by genetics and/or inadequate treatment response by the end of consolidation: Can the pEFS from time of randomization be improved by a treatment concept including two cycles of post-consolidation immunotherapy with blinatumomab (15 µg/m²/d for 28 days per cycle) plus 4 doses intrathecal Methotrexate replacing two conventional highly intensive chemotherapy courses?
- Randomization R-MR: Intermediate risk (MR) pB-ALL defined by genetics and intermediate MRD response: Can the probability of disease-free survival (pDFS) from time of randomization be improved by additional therapy with one cycle of post-reintensification immunotherapy with blinatumomab (15 µg/m²/d for 28 days)?
- Chapter 1.2.1: in the first paragraph: overall response 57%: patients not in CR can be added.
- Chapter 2.0: the efforts of current approaches with base edited CAR T to avoid fratricide for T-ALL should be mentioned too.
Author Response
Reviewer 1: Thank you for your comment, “The manuscript offers a broad overview on immunotherapy in pediatric ALL.” Please see our responses to your comments:
- Chapter 1.1.1: CRh, incomplete hematologic recovery should be defined; BLAST study: it should be specified that RFS is at 18 months
Response: The definition of CRh was added. For the BLAST study, the time point of 18 months was specified for the RFS observed.
- Table 1, at the bottom: Downs should be Down
Response: The edit was made as suggested.
- Chapter 1.1.2: at the beginning: the study reported (first in Table 1 and reference 14) should be defined as in the paper as “conducted in 26 European and US Centers”. The I-BFM is a consortium not restricted to Europe. Reference 15 at the end of this paragraph is not appropriate. Data published in 2016 (reference 14 could not be compared with those published in 2021 (reference 15).
Response: The I-BFM consortium was defined as suggested. Thank you for noting this reference error. The appropriate reference was cited.
- Chapter 1.1.2: the study in reference 15 is not a BFM and AIEOP study. It can be defined as an IntReALL study conducted in the frame of the I-BFM Study Group consortium. Please note that Munster should be Münster. In the same paragraph “3rd consolidation” should be “3rd consolidation course”.
Response: The edits to the study group title and the 3rd consolidation course were made as suggested.
- Chapter 1.1.2: at the end it can be added also that the “AIEOP-BFM ALL 2017 study, is conducting randomized studies in patients at intermediate and high risk EUDRACT Number 2016-001935-12; ClinicalTrials.gov Identifier: NCT03643276: see below
- Randomization R-HR: High-risk (HR) pB-ALL defined by genetics and/or inadequate treatment response by the end of consolidation: Can the pEFS from time of randomization be improved by a treatment concept including two cycles of post-consolidation immunotherapy with blinatumomab (15 µg/m²/d for 28 days per cycle) plus 4 doses intrathecal Methotrexate replacing two conventional highly intensive chemotherapy courses?
- Randomization R-MR: Intermediate risk (MR) pB-ALL defined by genetics and intermediate MRD response: Can the probability of disease-free survival (pDFS) from time of randomization be improved by additional therapy with one cycle of post-reintensification immunotherapy with blinatumomab (15 µg/m²/d for 28 days)?
Response: Thank you for this suggestion. The ongoing AIEOP-BFM ALL 2017 study was added to this section and to Table 1.
- Chapter 1.2.1: in the first paragraph: overall response 57%: patients not in CR can be added.
Response: The complete breakdown of the overall response rate was added.
- Chapter 2.0: the efforts of current approaches with base edited CAR T to avoid fratricide for T-ALL should be mentioned too.
Response: Current literature and studies addressing this area of CAR T was added.
Reviewer 2 Report
General review but all major points were well addressed.
Author Response
Reviewer 2: Thank you for your comment, “General review but all major points were well addressed.”
Reviewer 3 Report
The authors present a review of the current status and future of immunotherapy for childhood acute lymphoblastic leukemia.
The main focus of the paper is on CAR-T therapy, while results from adult patients are also presented for antibody therapy against CD19 or CD22.
CAR-T therapy for CD19 and CD22 is also discussed (as well as adult cases). Dual CAR-T for both is also mentioned, which is not too much.
The literature is well cited and this is a good, compact review article.
Author Response
Reviewer 3: Thank you for your comment, “The literature is well cited and this is a good, compact review article.”
Reviewer 4 Report
The review is well written and takes into account all published data of immunotherapy of pediatric ALL. Comparing pediatric data with data of adults underlines the value of the review.
Author Response
Reviewer 4: Thank you for your comment, “The review is well written and takes into account all published data of immunotherapy of pediatric ALL. Comparing pediatric data with data of adults underlines the value of the review.”